# Data Generation as Sequential Decision Making

**Philip Bachman**
McGill University, School of Computer Science
phil.bachman@gmail.com

**Doina Precup**
McGill University, School of Computer Science
dprecup@cs.mcgill.ca

## Abstract

We connect a broad class of generative models through their shared reliance on sequential decision making. Motivated by this view, we develop extensions to an existing model, and then explore the idea further in the context of data imputation – perhaps the simplest setting in which to investigate the relation between unconditional and conditional generative modelling. We formulate data imputation as an MDP and develop models capable of representing effective policies for it. We construct the models using neural networks and train them using a form of guided policy search [9]. Our models generate predictions through an iterative process of feedback and refinement. We show that this approach can learn effective policies for imputation problems of varying difficulty and across multiple datasets.

## 1 Introduction

Directed generative models are naturally interpreted as specifying sequential procedures for generating data. We traditionally think of this process as sampling, but one could also view it as making sequences of *decisions* for how to set the variables at each node in a model, conditioned on the settings of its parents, thereby generating data from the model. The large body of existing work on reinforcement learning provides powerful tools for addressing such sequential decision making problems. We encourage the use of these tools to understand and improve the extended processes currently driving advances in generative modelling. We show how sequential decision making can be applied to general prediction tasks by developing models which construct predictions by iteratively refining a working hypothesis under guidance from exogenous input and endogenous feedback.

We begin this paper by reinterpreting several recent generative models as sequential decision making processes, and then show how changes inspired by this point of view can improve the performance of the LSTM-based model introduced in [3]. Next, we explore the connections between directed generative models and reinforcement learning more fully by developing an approach to training policies for sequential data imputation. We base our approach on formulating imputation as a finite-horizon Markov Decision Process which one can also interpret as a deep, directed graphical model.

We propose two policy representations for the imputation MDP. One extends the model in [3] by inserting an explicit feedback loop into the generative process, and the other addresses the MDP more directly. We train our models/policies using techniques motivated by guided policy pearch [9, 10, 11, 8]. We examine their qualitative and quantitative performance across imputation problems covering a range of difficulties (i.e. different amounts of data to impute and different "missingness mechanisms"), and across multiple datasets. Given the relative paucity of existing approaches to the general imputation problem, we compare our models to each other and to two simple baselines. We also test how our policies perform when they use fewer/more steps to refine their predictions.

As imputation encompasses both classification and standard (i.e. unconditional) generative modelling, our work suggests that further study of models for the general imputation problem is worthwhile. The performance of our models suggests that sequential stochastic construction of predictions, guided by both input and feedback, should prove useful for a wide range of problems. Training these models can be challenging, but lessons from reinforcement learning may bring some relief.

## 2 Directed Generative Models as Sequential Decision Processes

Directed generative models have grown in popularity relative to their undirected counter-parts [6, 14, 12, 4, 5, 16, 15] (etc.). Reasons include: the development of efficient methods for training them, the ease of sampling from them, and the tractability of bounds on their log-likelihoods. Growth in available computing power compounds these benefits. One can interpret the (ancestral) sampling process in a directed model as repeatedly *setting* subsets of the latent variables to particular values, in a sequence of decisions conditioned on preceding decisions. Each subsequent decision restricts the set of potential outcomes for the overall sequence. Intuitively, these models encode stochastic procedures for constructing plausible observations. This section formally explores this perspective.

### 2.1 Deep AutoRegressive Networks

The deep autoregressive networks investigated in [4] define distributions of the following form:

$$p(x) = \sum_z p(x|z)p(z), \quad \text{with} \quad p(z) = p_0(z_0) \prod_{t=1}^{T} p_t(z_t|z_0, ..., z_{t-1}) \tag{1}$$

in which $x$ indicates a generated observation and $z_0, ..., z_T$ represent latent variables in the model. The distribution $p(x|z)$ may be factored similarly to $p(z)$. The form of $p(z)$ in Eqn. 1 can represent arbitrary distributions over the latent variables, and the work work in [4] mainly concerned approaches to parameterizing the conditionals $p_t(z_t|z_0, ..., z_{t-1})$ that restricted representational power in exchange for computational tractability. To appreciate the generality of Eqn. 1, consider using $z_t$ that are univariate, multivariate, structured, etc. One can interpret any model based on this sequential factorization of $p(z)$ as a non-stationary *policy* $p_t(z_t|s_t)$ for selecting each action $z_t$ in a state $s_t$, with each $s_t$ determined by all $z_{t'}$ for $t' < t$, and train it using some form of *policy search*.

### 2.2 Generalized Guided Policy Search

We adopt a broader interpretation of guided policy search than one might initially take from, e.g., [9, 10, 11, 8]. We provide a review of guided policy search in the supplementary material. Our expanded definition of guided policy search includes any optimization of the general form:

$$\underset{p,q}{\text{minimize}} \; \underset{i_q \sim \mathcal{I}_q}{\mathbb{E}} \; \underset{i_p \sim \mathcal{I}_p(\cdot|i_q)}{\mathbb{E}} \left[ \underset{\tau \sim q(\tau|i_q,i_p)}{\mathbb{E}} [\ell(\tau, i_q, i_p)] + \lambda \operatorname{div}(q(\tau|i_q, i_p), p(\tau|i_p)) \right] \tag{2}$$

in which $p$ indicates the *primary policy*, $q$ indicates the *guide policy*, $\mathcal{I}_q$ indicates a distribution over information available only to $q$, $\mathcal{I}_p$ indicates a distribution over information available to both $p$ and $q$, $\ell(\tau, i_q, i_p)$ computes the cost of trajectory $\tau$ in the context of $i_q/i_p$, and $\operatorname{div}(q(\tau|i_q, i_p), p(\tau|i_p))$ measures dissimilarity between the trajectory distributions generated by $p/q$. As $\lambda > 0$ goes to infinity, Eqn. 2 enforces the constraint $p(\tau|i_p) = q(\tau|i_q, i_p), \forall \tau, i_p, i_q$. Terms for controlling, e.g., the entropy of $p/q$ can also be added. The power of the objective in Eq. 2 stems from two main points: the guide policy $q$ can use information $i_q$ that is unavailable to the primary policy $p$, and the primary policy need only be trained to minimize the dissimilarity term $\operatorname{div}(q(\tau|i_q, i_p), p(\tau|i_p))$.

For example, a directed model structured as in Eqn. 1 can be interpreted as specifying a policy for a finite-horizon MDP whose terminal state distribution encodes $p(x)$. In this MDP, the state at time $1 \le t \le T+1$ is determined by $\{z_0, ..., z_{t-1}\}$. The policy picks an action $z_t \in \mathcal{Z}_t$ at time $1 \le t \le T$, and picks an action $x \in \mathcal{X}$ at time $t = T + 1$. I.e., the policy can be written as $p_t(z_t|z_0, ..., z_{t-1})$ for $1 \le t \le T$, and as $p(x|z_0, ..., z_T)$ for $t = T + 1$. The initial state $z_0 \in \mathcal{Z}_0$ is drawn from $p_0(z_0)$. Executing the policy for a single trial produces a trajectory $\tau \triangleq \{z_0, ..., z_T, x\}$, and the distribution over $x$s from these trajectories is just $p(x)$ in the corresponding directed generative model.

The authors of [4] train deep autoregressive networks by maximizing a variational lower bound on the training set log-likelihood. To do this, they introduce a variational distribution $q$ which provides $q_0(z_0|x^*)$ and $q_t(z_t|z_0, ..., z_{t-1}, x^*)$ for $1 \le t \le T$, with the final step $q(x|z_0, ..., z_T, x^*)$ given by a Dirac-delta at $x^*$. Given these definitions, the training in [4] can be interpreted as guided policy search for the MDP described in the previous paragraph. Specifically, the variational distribution $q$ provides a guide policy $q(\tau|x^*)$ over trajectories $\tau \triangleq \{z_0, ..., z_T, x^*\}$:

$$q(\tau|x^*) \triangleq q(x|z_0, ..., z_T, x^*)q_0(z_0|x^*) \prod_{t=1}^{T} q_t(z_t|z_0, ..., z_{t-1}, x^*) \tag{3}$$

The primary policy $p$ generates trajectories distributed according to:

$$p(\tau) \triangleq p(x|z_0, ..., z_T)p_0(z_0) \prod_{t=1}^{T} p_t(z_t|z_0, ..., z_{t-1}) \qquad (4)$$

which does not depend on $x^*$. In this case, $x^*$ corresponds to the guide-only information $i_q \sim \mathcal{I}_q$ in Eqn. 2. We now rewrite the variational optimization as:

$$\underset{p,q}{\text{minimize}} \underset{x^* \sim \mathcal{D}_{\mathcal{X}}}{\mathbb{E}} \left[ \underset{\tau \sim q(\tau|x^*)}{\mathbb{E}} [\ell(\tau, x^*)] + \text{KL}(q(\tau|x^*) \,||\, p(\tau)) \right] \qquad (5)$$

where $\ell(\tau, x^*) \triangleq 0$ and $\mathcal{D}_{\mathcal{X}}$ indicates the target distribution for the terminal state of the primary policy $p$.[1] When expanded, the KL term in Eqn. 5 becomes:

$$\text{KL}(q(\tau|x^*) \,||\, p(\tau)) = \qquad (6)$$

$$\underset{\tau \sim q(\tau|x^*)}{\mathbb{E}} \left[ \log \frac{q_0(z_0|x^*)}{p_0(z_0)} + \sum_{t=1}^{T} \log \frac{q_t(z_t|z_0, ..., z_{t-1}, x^*)}{p_t(z_t|z_0, ..., z_{t-1})} - \log p(x^*|z_0, ..., z_T) \right]$$

Thus, the variational approach used in [4] for training directed generative models can be interpreted as a form of generalized guided policy search. As the form in Eqn. 1 can represent any finite directed generative model, the preceding derivation extends to all models we discuss in this paper.[2]

## 2.3 Time-reversible Stochastic Processes

One can simplify Eqn. 1 by assuming suitable forms for $\mathcal{X}$ and $\mathcal{Z}_0, ..., \mathcal{Z}_T$. E.g., the authors of [16] proposed a model in which $\mathcal{Z}_t \equiv \mathcal{X}$ for all $t$ and $p_0(x_0)$ was Gaussian. We can write their model as:

$$p(x_T) = \sum_{x_0, ..., x_{T-1}} p_T(x_T|x_{T-1})p_0(x_0) \prod_{t=1}^{T-1} p_t(x_t|x_{t-1}) \qquad (7)$$

where $p(x_T)$ indicates the terminal state distribution of the non-stationary, finite-horizon Markov process determined by $\{p_0(x_0), p_1(x_1|x_0), ..., p_T(x_T|x_{T-1})\}$. Note that, throughout this paper, we (ab)use sums over latent variables and trajectories which could/should be written as integrals.

The authors of [16] observed that, for any reasonably smooth target distribution $\mathcal{D}_{\mathcal{X}}$ and sufficiently large $T$, one can define a "reverse-time" stochastic process $q_t(x_{t-1}|x_t)$ with simple, time-invariant dynamics that transforms $q(x_T) \triangleq \mathcal{D}_{\mathcal{X}}$ into the Gaussian distribution $p_0(x_0)$. This $q$ is given by:

$$q_0(x_0) = \sum_{x_1, ..., x_T} q_1(x_0|x_1)\mathcal{D}_{\mathcal{X}}(x_T) \prod_{t=2}^{T} q_t(x_{t-1}|x_t) \approx p_0(x_0) \qquad (8)$$

Next, we define $q(\tau)$ as the distribution over trajectories $\tau \triangleq \{x_0, ..., x_T\}$ generated by the reverse-time process determined by $\{q_1(x_0|x_1), ..., q_T(x_{T-1}|x_T), \mathcal{D}_{\mathcal{X}}(x_T)\}$. We define $p(\tau)$ as the distribution over trajectories generated by the "forward-time" process in Eqn. 7. The training in [16] is equivalent to guided policy search using guide trajectories sampled from $q$, i.e. it uses the objective:

$$\underset{p,q}{\text{minimize}} \underset{\tau \sim q(\tau)}{\mathbb{E}} \left[ \log \frac{q_1(x_0|x_1)}{p_0(x_0)} + \sum_{t=1}^{T-1} \log \frac{q_{t+1}(x_t|x_{t+1})}{p_t(x_t|x_{t-1})} + \log \frac{\mathcal{D}_{\mathcal{X}}(x_T)}{p_T(x_T|x_{T-1})} \right] \qquad (9)$$

which corresponds to minimizing $\text{KL}(q \,||\, p)$. If the log-densities in Eqn. 9 are tractable, then this minimization can be done using basic Monte-Carlo. If, as in [16], the reverse-time process $q$ is not trained, then Eqn. 9 simplifies to: $\text{minimize}_p \mathbb{E}_{q(\tau)} \left[ -\log p_0(x_0) - \sum_{t=1}^{T} \log p_t(x_t|x_{t-1}) \right]$.

This trick for generating guide trajectories exhibiting a particular distribution over terminal states $x_T$ – i.e. running dynamics backwards in time starting from $x_T \sim \mathcal{D}_{\mathcal{X}}$ – may prove useful in settings other than those considered in [16]. E.g., the LapGAN model in [1] learns to approximately invert a fixed (and information destroying) reverse-time process. The supplementary material expands on the content of this subsection, including a derivation of Eqn. 9 as a bound on $\mathbb{E}_{x \sim \mathcal{D}_{\mathcal{X}}}[-\log p(x)]$.

## 2.4 Learning Generative Stochastic Processes with LSTMs

The authors of [3] introduced a model for sequentially-deep generative processes. We interpret their model as a primary policy $p$ which generates trajectories $\tau \triangleq \{z_0, ..., z_T, x\}$ with distribution:

$$p(\tau) \triangleq p(x|s_\theta(\tau_{<x}))p_0(z_0)\prod_{t=1}^{T} p_t(z_t), \text{ with } \tau_{<x} \triangleq \{z_0, ..., z_T\} \tag{10}$$

in which $\tau_{<x}$ indicates a *latent trajectory* and $s_\theta(\tau_{<x})$ indicates a *state trajectory* $\{s_0, ..., s_T\}$ computed recursively from $\tau_{<x}$ using the update $s_t \leftarrow f_\theta(s_{t-1}, z_t)$ for $t \geq 1$. The initial state $s_0$ is given by a trainable constant. Each state $s_t \triangleq [h_t; v_t]$ represents the joint hidden/visible state $h_t/v_t$ of an LSTM and $f_\theta(\text{state}, \text{input})$ computes a standard LSTM update.[3] The authors of [3] defined all $p_t(z_t)$ as isotropic Gaussians and defined the output distribution $p(x|s_\theta(\tau_{<x}))$ as $p(x|c_T)$, where $c_T \triangleq c_0 + \sum_{t=1}^{T} \omega_\theta(v_t)$. Here, $c_0$ is a trainable constant and $\omega_\theta(v_t)$ is, e.g., an affine transform of $v_t$. Intuitively, $\omega_\theta(v_t)$ transforms $v_t$ into a refinement of the "working hypothesis" $c_{t-1}$, which gets updated to $c_t = c_{t-1} + \omega_\theta(v_t)$. $p$ is governed by parameters $\theta$ which affect $f_\theta$, $\omega_\theta$, $s_0$, and $c_0$. The supplementary material provides pseudo-code and an illustration for this model.

To train $p$, the authors of [3] introduced a guide policy $q$ with trajectory distribution:

$$q(\tau|x^*) \triangleq q(x|s_\phi(\tau_{<x}), x^*)q_0(z_0|x^*)\prod_{t=1}^{T} q_t(z_t|\tilde{s}_t, x^*), \text{ with } \tau_{<x} \triangleq \{z_0, ..., z_T\} \tag{11}$$

in which $s_\phi(\tau_{<x})$ indicates a state trajectory $\{\tilde{s}_0, ..., \tilde{s}_T\}$ computed recursively from $\tau_{<x}$ using the guide policy's state update $\tilde{s}_t \leftarrow f_\phi(\tilde{s}_{t-1}, g_\phi(s_\theta(\tau_{<t}), x^*))$. In this update $\tilde{s}_{t-1}$ is the previous guide state and $g_\phi(s_\theta(\tau_{<t}), x^*)$ is a deterministic function of $x^*$ and the partial (primary) state trajectory $s_\theta(\tau_{<t}) \triangleq \{s_0, ..., s_{t-1}\}$, which is computed recursively from $\tau_{<t} \triangleq \{z_0, ..., z_{t-1}\}$ using the state update $s_t \leftarrow f_\theta(s_{t-1}, z_t)$. The output distribution $q(x|s_\phi(\tau_{<x}), x^*)$ is defined as a Dirac-delta at $x^*$.[4] Each $q_t(z_t|\tilde{s}_t, x^*)$ is a diagonal Gaussian distribution with means and log-variances given by an affine function $L_\phi(\tilde{v}_t)$ of $\tilde{v}_t$. $q_0(z_0)$ is defined as identical to $p_0(z_0)$. $q$ is governed by parameters $\phi$ which affect the state updates $f_\phi(\tilde{s}_{t-1}, g_\phi(s_\theta(\tau_{<t}), x^*))$ and the step distributions $q_t(z_t|\tilde{s}_t, x^*)$. $g_\phi(s_\theta(\tau_{<t}), x^*)$ corresponds to the "read" operation of the encoder network in [3].

Using our definitions for $p/q$, the training objective in [3] is given by:

$$\underset{p,q}{\text{minimize}} \ \underset{x^*\sim\mathcal{D}_\mathcal{X}}{\mathbb{E}} \ \underset{\tau\sim q(\tau|x^*)}{\mathbb{E}} \left[ \sum_{t=1}^{T} \log \frac{q_t(z_t|\tilde{s}_t, x^*)}{p_t(z_t)} - \log p(x^*|s(\tau_{<x})) \right] \tag{12}$$

which can be written more succinctly as $\mathbb{E}_{x^*\sim\mathcal{D}_\mathcal{X}} \ \text{KL}(q(\tau|x^*) \,||\, p(\tau))$. This objective upper-bounds $\mathbb{E}_{x^*\sim\mathcal{D}_\mathcal{X}}[-\log p(x^*)]$, where $p(x) \triangleq \sum_{\tau_{<x}} p(x|s_\theta(\tau_{<x}))p(\tau_{<x})$.

## 2.5 Extending the LSTM-based Generative Model

We propose changing $p$ in Eqn. 10 to: $p(\tau) \triangleq p(x|s_\theta(\tau_{<x}))p_0(z_0)\prod_{t=1}^{T} p_t(z_t|s_{t-1})$. We define $p_t(z_t|s_{t-1})$ as a diagonal Gaussian distribution with means and log-variances given by an affine function $L_\theta(v_{t-1})$ of $v_{t-1}$ (remember that $s_t \triangleq [h_t; v_t]$), and we define $p_0(z_0)$ as an isotropic Gaussian. We set $s_0$ using $s_0 \leftarrow f_\theta(z_0)$, where $f_\theta$ is a trainable function (e.g. a neural network). Intuitively, our changes make the model more like a typical policy by conditioning its "action" $z_t$ on its state $s_{t-1}$, and upgrade the model to an infinite mixture by placing a distribution over its initial state $s_0$. We also consider using $c_t \triangleq L_\theta(h_t)$, which transforms the hidden part of the LSTM state $s_t$ directly into an observation. This makes $h_t$ a working memory in which to construct an observation. The supplementary material provides pseudo-code and an illustration for this model.

We train this model by optimizing the objective:

$$\underset{p,q}{\text{minimize}} \ \underset{x^*\sim\mathcal{D}_\mathcal{X}}{\mathbb{E}} \ \underset{\tau\sim q(\tau|x^*)}{\mathbb{E}} \left[ \log \frac{q_0(z_0|x^*)}{p_0(z_0)} + \sum_{t=1}^{T} \log \frac{q_t(z_t|\tilde{s}_t, x^*)}{p_t(z_t|s_{t-1})} - \log p(x^*|s(\tau_{<x})) \right] \tag{13}$$

where we now have to deal with $p_t(z_t|s_{t-1})$, $p_0(z_0)$, and $q_0(z_0|x^*)$, which could be treated as constants in the model from [3]. We define $q_0(z_0|x^*)$ as a diagonal Gaussian distribution whose means and log-variances are given by a trainable function $g_\phi(x^*)$.

When trained for the binarized MNIST benchmark used in [3], our extended model scored a negative log-likelihood of 85.5 on the test set.[5] For comparison, the score reported in [3] was 87.4.[6] After fine-tuning the variational distribution (i.e. $q$) on the test set, our model's score improved to 84.8, which is quite strong considering it is an upper bound. For comparison, see the best upper bound reported for this benchmark in [15], which was 85.1. When the model used the alternate $c_T \triangleq L_\theta(h_T)$, the raw/fine-tuned test scores were 85.9/85.3. Fig. 1 shows samples from the model. Model/test code is available at http://github.com/Philip-Bachman/Sequential-Generation.

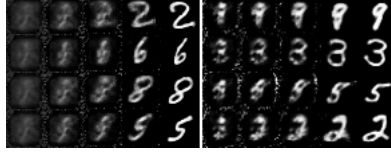

Figure 1: The left block shows $\sigma(c_t)$ for $t \in \{1, 3, 5, 9, 16\}$, for a policy $p$ with $c_t \triangleq c_0 + \sum_{t'=1}^{t} L_\theta(v_{t'})$. The right block is analogous, for a model using $c_t \triangleq L_\theta(h_t)$.

## 3 Developing Models for Sequential Imputation

The goal of imputation is to estimate $p(x^u|x^k)$, where $x \triangleq [x^u; x^k]$ indicates a *complete observation* with *known values* $x^k$ and *missing values* $x^u$. We define a *mask* $m \in \mathcal{M}$ as a (disjoint) partition of $x$ into $x^u/x^k$. By expanding $x^u$ to include all of $x$, one recovers standard generative modelling. By shrinking $x^u$ to include a single element of $x$, one recovers standard classification/regression. Given distribution $\mathcal{D}_\mathcal{M}$ over $m \in \mathcal{M}$ and distribution $\mathcal{D}_\mathcal{X}$ over $x \in \mathcal{X}$, the objective for imputation is:

$$\underset{p}{\text{minimize}} \; \underset{x \sim \mathcal{D}_\mathcal{X}}{\mathbb{E}} \; \underset{m \sim \mathcal{D}_\mathcal{M}}{\mathbb{E}} \left[ -\log p(x^u|x^k) \right] \qquad (14)$$

We now describe a finite-horizon MDP for which guided policy search minimizes a bound on the objective in Eqn. 14. The MDP is defined by mask distribution $\mathcal{D}_\mathcal{M}$, complete observation distribution $\mathcal{D}_\mathcal{X}$, and the state spaces $\{\mathcal{Z}_0, ..., \mathcal{Z}_T\}$ associated with each of $T$ steps. Together, $\mathcal{D}_\mathcal{M}$ and $\mathcal{D}_\mathcal{X}$ define a joint distribution over initial states and rewards in the MDP. For the trial determined by $x \sim \mathcal{D}_\mathcal{X}$ and $m \sim \mathcal{D}_\mathcal{M}$, the initial state $z_0 \sim p(z_0|x^k)$ is selected by the policy $p$ based on the known values $x^k$. The cost $\ell(\tau, x^u, x^k)$ suffered by trajectory $\tau \triangleq \{z_0, ..., z_T\}$ in the context $(x, m)$ is given by $-\log p(x^u|\tau, x^k)$, i.e. the negative log-likelihood of $p$ guessing the missing values $x^u$ after following trajectory $\tau$, while seeing the known values $x^k$.

We consider a policy $p$ with trajectory distribution $p(\tau|x^k) \triangleq p(z_0|x^k) \prod_{t=1}^{T} p(z_t|z_0, ..., z_{t-1}, x^k)$, where $x^k$ is determined by $x/m$ for the current trial and $p$ can't observe the missing values $x^u$. With these definitions, we can find an approximately optimal imputation policy by solving:

$$\underset{p}{\text{minimize}} \; \underset{x \sim \mathcal{D}_\mathcal{X}}{\mathbb{E}} \; \underset{m \sim \mathcal{D}_\mathcal{M}}{\mathbb{E}} \; \underset{\tau \sim p(\tau|x^k)}{\mathbb{E}} \left[ -\log p(x^u|\tau, x^k) \right] \qquad (15)$$

I.e. the expected negative log-likelihood of making a correct imputation on any given trial. This is a valid, but loose, upper bound on the imputation objective in Eq. 14 (from Jensen's inequality). We can tighten the bound by introducing a guide policy (i.e. a variational distribution).

As with the unconditional generative models in Sec. 2, we train $p$ to imitate a guide policy $q$ shaped by additional information (here it's $x^u$). This $q$ generates trajectories with distribution $q(\tau|x^u, x^k) \triangleq q(z_0|x^u, x^k) \prod_{t=1}^{T} q(z_t|z_0, ..., z_{t-1}, x^u, x^k)$. Given this $p$ and $q$, guided policy search solves:

$$\underset{p,q}{\text{minimize}} \; \underset{x \sim \mathcal{D}_\mathcal{X}}{\mathbb{E}} \; \underset{m \sim \mathcal{D}_\mathcal{M}}{\mathbb{E}} \left[ \underset{\tau \sim q(\tau|i_q, i_p)}{\mathbb{E}} \left[ -\log q(x^u|\tau, i_q, i_p) \right] + \text{KL}(q(\tau|i_q, i_p) \, || \, p(\tau|i_p)) \right] \qquad (16)$$

where we define $i_q \triangleq x^u$, $i_p \triangleq x^k$, and $q(x^u|\tau, i_q, i_p) \triangleq p(x^u|\tau, i_p)$.

[5]Data splits from: http://www.cs.toronto.edu/~larocheh/public/datasets/binarized_mnist
[6]The model in [3] significantly improves its score to 80.97 when using an image-specific architecture.

## 3.1 A Direct Representation for Sequential Imputation Policies

We define an imputation trajectory as $c_\tau \triangleq \{c_0, ..., c_T\}$, where each partial imputation $c_t \in \mathcal{X}$ is computed from a partial step trajectory $\tau_{<t} \triangleq \{z_1, ..., z_t\}$. A partial imputation $c_{t-1}$ encodes the policy's guess for the missing values $x^u$ immediately prior to selecting step $z_t$, and $c_T$ gives the policy's final guess. At each step of iterative refinement, the policy selects a $z_t$ based on $c_{t-1}$ and the known values $x^k$, and then updates its guesses to $c_t$ based on $c_{t-1}$ and $z_t$. By iteratively refining its guesses based on feedback from earlier guesses and the known values, the policy can construct complexly structured distributions over its final guess $c_T$ after just a few steps. This happens naturally, without any post-hoc MRFs/CRFs (as in many approaches to structured prediction), and without sampling values in $c_T$ one at a time (as required by existing NADE-type models [7]). This property of our approach should prove useful for many tasks.

We consider two ways of updating the guesses in $c_t$, mirroring those described in Sec. 2. The first way sets $c_t \leftarrow c_{t-1} + \omega_\theta(z_t)$, where $\omega_\theta(z_t)$ is a trainable function. We set $c_0 \triangleq [c_0^u; c_0^k]$ using a trainable bias. The second way sets $c_t \leftarrow \omega_\theta(z_t)$. We indicate models using the first type of update with the suffix *-add*, and models using the second type of update with *-jump*. Our primary policy $p_\theta$ selects $z_t$ at each step $1 \le t \le T$ using $p_\theta(z_t|c_{t-1}, x^k)$, which we restrict to be a diagonal Gaussian. This is a simple, stationary policy. Together, the step selector $p_\theta(z_t|c_{t-1}, x^k)$ and the imputation constructor $\omega_\theta(z_t)$ fully determine the behaviour of the primary policy. The supplementary material provides pseudo-code and an illustration for this model.

We construct a guide policy $q$ similarly to $p$. The guide policy shares the imputation constructor $\omega_\theta(z_t)$ with the primary policy. The guide policy incorporates additional information $x \triangleq [x^u; x^k]$, i.e. the complete observation for which the primary policy must reconstruct some missing values. The guide policy chooses steps using $q_\phi(z_t|c_{t-1}, x)$, which we restrict to be a diagonal Gaussian.

We train the primary/guide policy components $\omega_\theta$, $p_\theta$, and $q_\phi$ simultaneously on the objective:

$$\underset{\theta, \phi}{\text{minimize}} \;\; \underset{x \sim \mathcal{D}_\mathcal{X}}{\mathbb{E}} \; \underset{m \sim \mathcal{D}_\mathcal{M}}{\mathbb{E}} \left[ \underset{\tau \sim q_\phi(\tau|x^u, x^k)}{\mathbb{E}} [-\log q(x^u|c_T^u)] + \text{KL}(q(\tau|x^u, x^k) \,||\, p(\tau|x^k)) \right] \quad (17)$$

where $q(x^u|c_T^u) \triangleq p(x^u|c_T^u)$. We train our models using Monte-Carlo roll-outs of $q$, and stochastic backpropagation as in [6, 14]. Full implementations and test code are available from `http://github.com/Philip-Bachman/Sequential-Generation`.

## 3.2 Representing Sequential Imputation Policies using LSTMs

To make it useful for imputation, which requires conditioning on the exogenous information $x^k$, we modify the LSTM-based model from Sec. 2.5 to include a "read" operation in its primary policy $p$. We incorporate a read operation by spreading $p$ over two LSTMs, $p^r$ and $p^w$, which respectively "read" and "write" an imputation trajectory $c_\tau \triangleq \{c_0, ..., c_T\}$. Conveniently, the guide policy $q$ for this model takes the same form as the primary policy's reader $p^r$. This model also includes an "infinite mixture" initialization step, as used in Sec. 2.5, but modified to incorporate conditioning on $x$ and $m$. The supplementary material provides pseudo-code and an illustration for this model.

Following the infinite mixture initialization step, a single full step of execution for $p$ involves several substeps: first $p$ updates the reader state using $s_t^r \leftarrow f_\theta^r(s_{t-1}^r, \omega_\theta^r(c_{t-1}, s_{t-1}^w, x^k))$, then $p$ selects a step $z_t \sim p_\theta(z_t|v_t^r)$, then $p$ updates the writer state using $s_t^w \leftarrow f_\theta^w(s_{t-1}^w, z_t)$, and finally $p$ updates its guesses by setting $c_t \leftarrow c_{t-1} + \omega_\theta^w(v_t^w)$ (or $c_t \leftarrow \omega_\theta^w(h_t^w)$). In these updates, $s_t^{r,w} \triangleq [h_t^{r,w}; v_t^{r,w}]$ refer to the states of the $(r)$reader and $(w)$writer LSTMs. The LSTM updates $f_\theta^{r,w}$ and the read/write operations $\omega_\theta^{r,w}$ are governed by the policy parameters $\theta$.

We train $p$ to imitate trajectories sampled from a guide policy $q$. The guide policy shares the primary policy's writer updates $f_\theta^w$ and write operation $\omega_\theta^w$, but has its own reader updates $f_\phi^q$ and read operation $\omega_\phi^q$. At each step, the guide policy: updates the guide state $s_t^q \leftarrow f_\phi^q(s_{t-1}^q, \omega_\phi^q(c_{t-1}, s_{t-1}^w, x))$, then selects $z_t \sim q_\phi(z_t|v_t^q)$, then updates the writer state $s_t^w \leftarrow f_\theta^w(s_{t-1}^w, z_t)$, and finally updates its guesses $c_t \leftarrow c_{t-1} + \omega_\theta^w(v_t^w)$ (or $c_t \leftarrow \omega_\theta^w(h_t^w)$). As in Sec. 3.1, the guide policy's read operation $\omega_\phi^q$ gets to see the *complete observation* $x$, while the primary policy only gets to see the known values $x^k$. We restrict the step distributions $p_\theta/q_\phi$ to be diagonal Gaussians whose means and log-variances are affine functions of $v_t^r/v_t^q$. The training objective has the same form as Eq. 17.

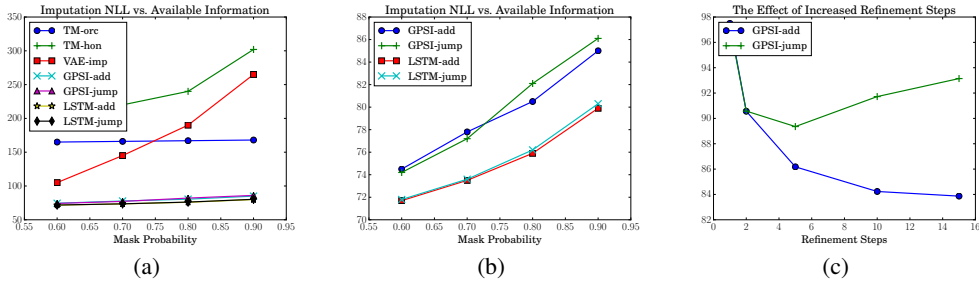

(a)                  (b)                  (c)

Figure 2: (a) Comparing the performance of our imputation models against several baselines, using MNIST digits. The $x$-axis indicates the % of pixels which were dropped completely at random, and the scores are normalized by the number of imputed pixels. (b) A closer view of results from (a), just for our models. (c) The effect of increased iterative refinement steps for our GPSI models.

## 4 Experiments

We tested the performance of our sequential imputation models on three datasets: MNIST (28x28), SVHN (cropped, 32x32) [13], and TFD (48x48) [17]. We converted images to grayscale and shift/scaled them to be in the range [0...1] prior to training/testing. We measured the imputation log-likelihood $\log q(x^u|c_T^u)$ using the true missing values $x^u$ and the models' guesses given by $\sigma(c_T^u)$. We report negative log-likelihoods, so lower scores are better in all of our tests. We refer to variants of the model from Sec. 3.1 as GPSI-add and GPSI-jump, and to variants of the model from Sec. 3.2 as LSTM-add and LSTM-jump. Except where noted, the GPSI models used 6 refinement steps and the LSTM models used 16.[7]

We tested imputation under two types of data masking: missing completely at random (MCAR) and missing at random (MAR). In MCAR, we masked pixels uniformly at random from the source images, and indicate removal of $d$% of the pixels by MCAR-$d$. In MAR, we masked square regions, with the occlusions located uniformly at random within the borders of the source image. We indicate occlusion of a $d \times d$ square by MAR-$d$.

On MNIST, we tested MCAR-$d$ for $d \in \{50, 60, 70, 80, 90\}$. MCAR-100 corresponds to unconditional generation. On TFD and SVHN we tested MCAR-80. On MNIST, we tested MAR-$d$ for $d \in \{14, 16\}$. On TFD we tested MAR-25 and on SVHN we tested MAR-17. For test trials we sampled masks from the same distribution used in training, and we sampled complete observations from a held-out test set. Fig. 2 and Tab. 1 present quantitative results from these tests. Fig. 2(c) shows the behavior of our GPSI models when we allowed them fewer/more refinement steps.

|  | MNIST | | TFD | | SVHN | |
|---|---|---|---|---|---|---|
|  | MAR-14 | MAR-16 | MCAR-80 | MAR-25 | MCAR-80 | MAR-17 |
| LSTM-add | 170 | 167 | 1381 | 1377 | 525 | 568 |
| LSTM-jump | 172 | 169 | – | – | – | – |
| GPSI-add | 177 | 175 | 1390 | 1380 | 531 | 569 |
| GPSI-jump | 183 | 177 | 1394 | 1384 | 540 | 572 |
| VAE-imp | 374 | 394 | 1416 | 1399 | 567 | 624 |

Table 1: Imputation performance in various settings. Details of the tests are provided in the main text. Lower scores are better. Due to time constraints, we did not test LSTM-jump on TFD or SVHN. These scores are normalized for the number of imputed pixels.

We tested our models against three baselines. The baselines were "variational auto-encoder imputation", honest template matching, and oracular template matching. VAE imputation ran multiple steps of VAE reconstruction, with the known values held fixed and the missing values re-estimated with each reconstruction step.[8] After 16 refinement steps, we scored the VAE based on its best

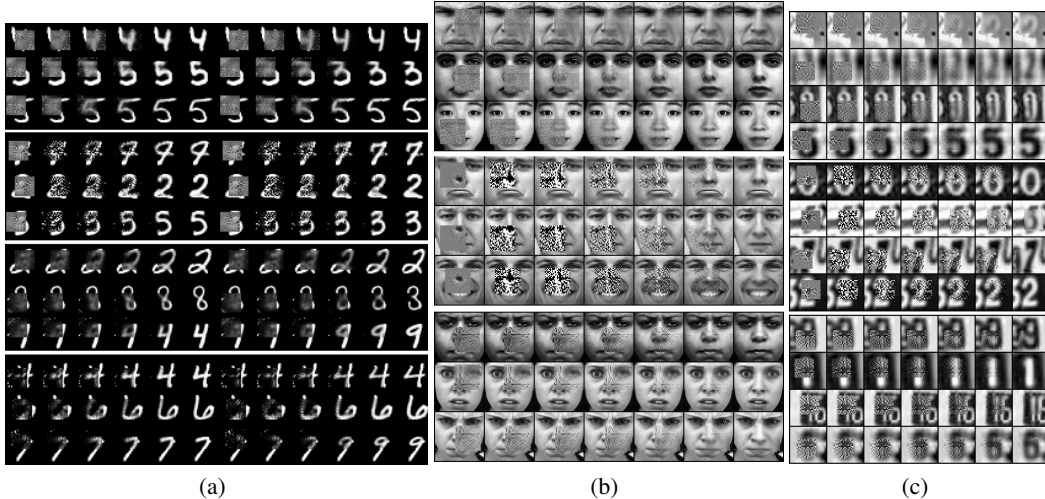

<div align="center">(a)                  (b)                  (c)</div>

Figure 3: This figure illustrates the policies learned by our models. (a): models trained for (MNIST, MAR-16). From top→bottom the models are: GPSI-add, GPSI-jump, LSTM-add, LSTM-jump. (b): models trained for (TFD, MAR-25), with models in the same order as (a) – but without LSTM-jump. (c): models trained for (SVHN, MAR-17), with models arranged as for (b).

guesses. Honest template matching guessed the missing values based on the training image which best matched the test image's known values. Oracular template matching was like honest template matching, but matched directly on the missing values.

Our models significantly outperformed the baselines. In general, the LSTM-based models outperformed the more direct GPSI models. We evaluated the log-likelihood of imputations produced by our models using the lower bounds provided by the variational objectives with respect to which they were trained. Evaluating the template-based imputations was straightforward. For VAE imputation, we used the expected log-likelihood of the imputations sampled from multiple runs of the 16-step imputation process. This provides a valid, but loose, lower bound on their log-likelihood.

As shown in Fig. 3, the imputations produced by our models appear promising. The imputations are generally of high quality, and the models are capable of capturing strongly multi-modal reconstruction distributions (see subfigure (a)). The behavior of GPSI models changed intriguingly when we swapped the imputation constructor. Using the *-jump* imputation constructor, the imputation policy learned by the direct model was rather inscrutable. Fig. 2(c) shows that additive guess updates extracted more value from using more refinement steps. When trained on the binarized MNIST benchmark discussed in Sec. 2.5, i.e. with binarized images and subject to MCAR-100, the LSTM-add model produced raw/fine-tuned scores of 86.2/85.7. The LSTM-jump model scored 87.1/86.3. Anecdotally, on this task, these "closed-loop" models seemed more prone to overfitting than the "open-loop" models in Sec. 2.5. The supplementary material provides further qualitative results.

## 5 Discussion

We presented a point of view which links methods for training directed generative models with policy search in reinforcement learning. We showed how our perspective can guide improvements to existing models. The importance of these connections will only grow as generative models rapidly increase in structural complexity and effective decision depth.

We introduced the notion of imputation as a natural generalization of standard, unconditional generative modelling. Depending on the relation between the data-to-generate and the available information, imputation spans from full unconditional generative modelling to classification/regression. We showed how to successfully train sequential imputation policies comprising millions of parameters using an approach based on guided policy search [9]. Our approach outperforms the baselines quantitatively and appears qualitatively promising. Incorporating, e.g., the local read/write mechanisms from [3] should provide further improvements.

## Footnotes

[1] We could pull the $-\log p(x^*|z_0, ..., z_T)$ term from the KL and put it in the cost $\ell(\tau, x^*)$, but we prefer the "path-wise KL" formulation for its elegance. We abuse notation using $\text{KL}(\delta(x = x^*) \,||\, p(x)) \triangleq -\log p(x^*)$.

[2] This also includes all generative models implemented and executed on an actual computer.

[3]For those unfamiliar with LSTMs, a good introduction can be found in [2]. We use LSTMs including input gates, forget gates, output gates, and peephole connections for all tests presented in this chapter.

[4]It may be useful to relax this assumption.

[7]GPSI stands for "Guided Policy Search Imputer". The tag "-add" refers to additive guess updates, and "-jump" refers to updates that fully replace the guesses.

[8]We discuss some deficiencies of VAE imputation in the supplementary material.

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
