[Supplementary Material]

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

# 6 Appendix

# 7 Additional Material for Section 2

## 7.1 A Brief Review of Policy Search and Guided Policy Search

Policy search refers to a general class of methods for searching directly through the space of possible parameterized policies for a reinforcement learning system (in contrast to fitting a value function and determining the policy implicitly by choosing the best actions). However, policy search is subject to local optima, which can be quite bad if the policy space is very rich (e.g., policies represented by deep networks). Guided policy search methods [11, 12, 13, 10] address this problem by using either guiding samples, or a guide policy (which generates guiding samples), in order to help move the policy search away from bad local optima. We refer to "local optima" in a colloquial/practical sense. I.e. regions of policy space in which the policy is unlikely to improve via noisy local search.

The initial approach to this problem was to generate guiding samples from policies obtained through trajectory optimization using differential dynamic programming [11]. After applying importance sampling corrections, the guiding samples were then used for off-policy training of the primary policy, a standard approach in policy search. Further work has obtained samples by using a "guide policy" which typically belongs to a larger policy class than the one being searched [13, 10]. In both cases, the optimization criterion contains, in addition to the reward, a regularization term requiring trajectories from the trained policy to be close to the guide samples. Constraining divergence between the guide samples and the trajectories produced by the trained policy allows the system generating the guide samples to gradually pull the trained policy towards improved behavior.

## 7.2 A Path-wise $\mathrm{KL}$ Bound for Reversible Stochastic Processes

We now show that the objective in Eqn. 9 describes the KL divergence $\mathrm{KL}(q_\tau \,\|\, p_\tau)$, and that it provides an upper bound on $\mathbb{E}_{\mathcal{D}_\mathcal{X}}[-\log p(x_T)]$. First, for $\tau \triangleq \{x_0, ..., x_T\}$, we define:

- $p(\tau_{>0}|x_0) \triangleq p(x_1, ..., x_T|x_0) \triangleq \prod_{t=1}^{T} p_t(x_t|x_{t-1})$

- $p(\tau) \triangleq p(x_1, ..., x_T|x_0)p_0(x_0) \triangleq p_0(x_0) \prod_{t=1}^{T} p_t(x_t|x_{t-1})$

- $q(\tau_{<T}|x_T) \triangleq q(x_0, ..., x_{T-1}|x_T) \triangleq \prod_{t=1}^{T} q_t(x_{t-1}|x_t)$

- $q(\tau) \triangleq q(x_0, ..., x_{T-1}|x_T)\mathcal{D}_\mathcal{X}(x_T) \triangleq \mathcal{D}_\mathcal{X}(x_T) \prod_{t=1}^{T} q_t(x_{t-1}|x_t)$

Next, we derive:

$$
p(x_T) = \sum_{x_0,...,x_{T-1}} p_0(x_0)p(x_1,...,x_T|x_0)\frac{q(\tau_{<T}|x_T)}{q(\tau_{<T}|x_T)} \tag{18}
$$

$$
= \sum_{x_0,...,x_{T-1}} p_0(x_0)p(\tau_{>0}|x_0)\frac{q(\tau_{<T}|x_T)}{q(\tau_{<T}|x_T)} \tag{19}
$$

$$
= \sum_{x_0,...,x_{T-1}} q(\tau_{<T}|x_T)\frac{p_0(x_0)p(\tau_{>0}|x_0)}{q(\tau_{<T}|x_T)} \tag{20}
$$

$$
= \sum_{x_0,...,x_{T-1}} q(x_0,...,x_{T-1}|x_T) \cdot \left( p_0(x_0)\prod_{t=1}^{T}\frac{p_t(x_t|x_{t-1})}{q_t(x_{t-1}|x_t)} \right) \tag{21}
$$

$$
\log p(x_T) \geq \sum_{x_0,...,x_{T-1}} q(x_0,...,x_{T-1}|x_T) \cdot \log \left( p_0(x_0)\prod_{t=1}^{T}\frac{p_t(x_t|x_{t-1})}{q_t(x_{t-1}|x_t)} \right) \tag{22}
$$

$$
\geq \mathop{\mathbb{E}}_{q(\tau_{<T}|x_T)} \left[ \log p_0(x_0) - \log\prod_{t=1}^{T}\frac{q_t(x_{t-1}|x_t)}{p_t(x_t|x_{t-1})} \right] \tag{23}
$$

$$
\geq \mathop{\mathbb{E}}_{q(\tau_{<T}|x_T)} \left[ \log p_0(x_0) - \log\frac{q(\tau_{<T}|x_T)}{p(\tau_{>0}|x_0)} \right] \tag{24}
$$

$$
\geq \mathop{\mathbb{E}}_{q(\tau_{<T}|x_T)} \left[ \log p_0(x_0) \right] - \mathrm{KL}(q(\tau_{<T}|x_T)\,||\,p(\tau_{>0}|x_0)) \tag{25}
$$

which provides a lower bound on $\log p(x_T)$ based on sample trajectories produced by the reverse-time process $q$ when it is started at $x_T$. The transition from equality to inequality is due to Jensen's inequality. Though $q(\tau_{<T}|x_T)$ and $p(\tau_{>0}|x_0)$ may at first seem incommensurable via KL, they both represent distributions over $T$-step trajectories through $\mathcal{X}$ space, and thus the required KL divergence is well-defined. Next, by adding an expectation with respect to $x_T \sim \mathcal{D}_\mathcal{X}$, we derive a lower bound on the expected log-likelihood $\mathbb{E}_{\mathcal{D}_\mathcal{X}}[\log p(x_T)]$:

$$
\log p(x_T) \geq \mathop{\mathbb{E}}_{q(\tau_{<T}|x_T)} \left[ \log p_0(x_0) - \log\frac{q(\tau_{<T}|x_T)}{p(\tau_{>0}|x_0)} \right] \tag{26}
$$

$$
\mathop{\mathbb{E}}_{x_T\sim\mathcal{D}_\mathcal{X}}[\log p(x_T)] \geq \mathop{\mathbb{E}}_{x_T\sim\mathcal{D}_\mathcal{X}} \left[ \mathop{\mathbb{E}}_{q(\tau_{<T}|x_T)} \left[ \log p_0(x_0) - \log\frac{q(\tau_{<T}|x_T)}{p(\tau_{>0}|x_0)} \right] \right] \tag{27}
$$

$$
\geq \mathop{\mathbb{E}}_{q(\tau)} \left[ \log p_0(x_0) - \log\frac{q(\tau_{<T}|x_T)}{p(\tau_{>0}|x_0)} \right] \tag{28}
$$

$$
\geq \mathop{\mathbb{E}}_{q(\tau)} \left[ -\log\frac{\mathcal{D}(x_T)q(\tau_{<T}|x_T)}{p_0(x_0)p(\tau_{>0}|x_0)} \right] - H_{\mathcal{D}_\mathcal{X}} \tag{29}
$$

$$
\geq -\mathrm{KL}(q(\tau)\,||\,p(\tau)) - H_{\mathcal{D}_\mathcal{X}} \tag{30}
$$

These steps follow directly from the definitions of $q(\tau_{<T}|x_T)$ and $q(\tau)$. In the last two equations, we define $H_{\mathcal{D}_\mathcal{X}} \triangleq \mathbb{E}_{x\sim\mathcal{D}_\mathcal{X}}[-\log\mathcal{D}_\mathcal{X}(x)]$, which gives the entropy of $\mathcal{D}_\mathcal{X}$. Thus, when $\mathcal{D}_\mathcal{X}$ is constant with respect to the trainable parameters, the training objective in [18] is equivalent to minimizing the path-based $\mathrm{KL}(q(\tau)\,||\,p(\tau))$.

# 8   Additional material for Section 3

**The LSTM-based generative model from Section 2.4**

Figure 4: **Left:** this figure illustrates the structure of the LSTM-based model from [4], as described in Sec. 2.4. Single-edged nodes are deterministic and double-edged nodes are stochastic. Dashed nodes and edges are present only during training. **Right:** this figure provides pseudo-code for the loop that computes all values required for computing this model's training objective. The objective follows the form of Eqn. 5. To simplify notation, we don't distinguish between the visible/hidden states of the LSTMs.

**Algorithm 1** GenTrainLoop1( $x^*$ )

1: Set $s_0$, $\tilde{s}_0$, and $c_0$ from constants.
2: Compute $\text{nll}_0 = -\log p(x^*|c_0)$.
3: Set $\text{kl}_0$ to 0.
4: **for** $t = 1$ to $T$ **do**
5:     Update $\tilde{s}_t \leftarrow f_\phi(\tilde{s}_{t-1}, g_\phi(s_{t-1}, c_{t-1}, x^*))$.
6:     Sample $z_t \sim q_\phi(z_t|\tilde{s}_t)$.
7:     Update $s_t \leftarrow f_\theta(s_{t-1}, z_t)$.
8:     Update $c_t \leftarrow c_{t-1} + \omega_\theta(s_t)$ (or $\omega_\theta(s_t)$).
9:     Compute $\text{kl}_t = \text{KL}(q_\phi(z_t|\tilde{s}_t) \,||\, p_\theta(z_t))$.
10:     Compute $\text{nll}_t = -\log p(x^*|c_t)$.
11: **end for**
12: **return** $c_{0:T}$, $\text{nll}_{0:T}$, and $\text{kl}_{0:T}$.

**The extended LSTM-based generative model from Section 2.5**

Figure 5: **Left:** this figure illustrates the structure of the extended LSTM-based model described in Sec. 2.5. Single-edged nodes are deterministic and double-edged nodes are stochastic. Dashed nodes and edges are present only during training. **Right:** this figure provides pseudo-code for the loop that computes all values required for computing this model's training objective. The objective follows the form of Eqn. 5. To simplify notation, we don't distinguish between the visible/hidden states of the LSTMs.

**Algorithm 1** GenTrainLoop2( $x^*$ )

1: Sample $z_0 \sim q_\phi(z_0|x^*)$.
2: Set $s_0$ and $\tilde{s}_0$ from $f_\theta(z_0)$.
3: Set $c_0$ from a constant.
4: Compute $\text{kl}_0 = \text{KL}(q_\phi(z_0|x^*) \,||\, p_\theta(z_0))$.
5: Compute $\text{nll}_0 = -\log p(x^*|c_0)$.
6: **for** $t = 1$ to $T$ **do**
7:     Update $\tilde{s}_t \leftarrow f_\phi(\tilde{s}_{t-1}, g_\phi(s_{t-1}, c_{t-1}, x^*))$.
8:     Sample $z_t \sim q_\phi(z_t|\tilde{s}_t)$.
9:     Update $s_t \leftarrow f_\theta(s_{t-1}, z_t)$.
10:     Update $c_t \leftarrow c_{t-1} + \omega_\theta(s_t)$ (or $\omega_\theta(s_t)$).
11:     Compute $\text{kl}_t = \text{KL}(q_\phi(z_t|\tilde{s}_t) \,||\, p_\theta(z_t|z_{t-1}))$.
12:     Compute $\text{nll}_t = -\log p(x^*|c_t)$.
13: **end for**
14: **return** $c_{0:T}$, $\text{nll}_{0:T}$, and $\text{kl}_{0:T}$.

## The direct imputation model from Section 3.1

**Algorithm 1** ImpTrainLoop1( $x, m$ )

1: Set $x^k, x^u \leftarrow$ ApplyMask$(x, m)$.
2: Set $c_0$ from a constant.
3: Set $\text{nll}_0$ and $\text{kl}_0$ to 0.
4: **for** $t = 1$ to $T$ **do**
5:      Sample $z_t \sim q_\phi(z_t|c_{t-1}, x^k, x^u)$.
6:      Update $c_t \leftarrow c_{t-1} + \omega_\theta(z_t)$ (or $\omega_\theta(z_t)$).
7:      Compute $\text{kl}_t = \text{KL}(q_\phi(z_t|c_{t-1}, x^k, x^u) \,||\, p_\theta(z_t|c_{t-1}, x^k))$.
8:      Compute $\text{nll}_t = -\log p(x^u|c_t)$.
9: **end for**
10: **return** $c_{0:T}$, $\text{nll}_{0:T}$, and $\text{kl}_{0:T}$.

Figure 6: **Left:** this figure illustrates the structure of the "direct" imputation model described in Sec. 3.1. Single-edged nodes are deterministic and double-edged nodes are stochastic. All solid lines affect the primary and guide policies. All dashed lines affect only the guide policy. **Right:** this figure provides pseudo-code for the loop that computes all values required for computing this model's training objective. The objective follows the form of Eqn. 17. To simplify notation, we don't distinguish between the visible/hidden states of the LSTMs.

## The LSTM-based imputation model Section 3.2

**Algorithm 1** ImpTrainLoop2( $x, m$ )

1: Set $x^k, x^u \leftarrow$ ApplyMask$(x, m)$.
2: Set $c_0$ from a constant.
3: Sample $z_0 \sim q_\phi(z_0|c_0, x^k, x^u)$.
4: Set $s_0^r$, $s_0^w$, and $s_0^q$ from $f_\theta(z_0)$.
5: Compute $\text{kl}_0 = \text{KL}(q_\phi(z_0|c_0, x^k, x^u) \,||\, p_\theta(z_0|c_0, x^k))$.
6: Compute $\text{nll}_0 = -\log p(x^u|c_0)$.
7: **for** $t = 1$ to $T$ **do**
8:      Update $s_t^q \leftarrow f_\phi(s_{t-1}^q, \omega_\phi^q(s_{t-1}^q, c_{t-1}, x^k, x^u))$.
9:      Update $s_t^r \leftarrow f_\theta(s_{t-1}^r, \omega_\theta^r(s_{t-1}^r, c_{t-1}, x^k))$.
10:      Sample $z_t \sim q_\phi(z_t|s_t^q)$.
11:      Update $s_t^w \leftarrow f_\theta(s_{t-1}^w, z_t)$.
12:      Update $c_t \leftarrow c_{t-1} + \omega_\theta^w(s_t^w)$ (or $\omega_\theta^w(s_t^w)$).
13:      Compute $\text{kl}_t = \text{KL}(q_\phi(z_t|s_t^q) \,||\, p_\theta(z_t|s_t^r))$.
14:      Compute $\text{nll}_t = -\log p(x^u|c_t)$.
15: **end for**
16: **return** $c_{0:T}$, $\text{nll}_{0:T}$, and $\text{kl}_{0:T}$.

Figure 7: **Left:** this figure illustrates the structure of the "LSTM" imputation model described in Sec. 3.2. Single-edged nodes are deterministic and double-edged nodes are stochastic. All solid lines affect the primary and guide policies. All dashed lines affect only the guide policy. **Right:** this figure provides pseudo-code for the loop that computes all values required for computing this model's training objective. The objective follows the form of Eqn. 17. To simplify notation, we don't distinguish between the visible/hidden states of the LSTMs.

# 9 Additional Material for Experiments and Model Implementations

## 9.1 Model Implementation Details

For purely generative tests, all LSTMs had hidden and visible states in $\mathbb{R}^{250}$. We ran the LSTMs for 16 steps. For our extended model in Sec. 2.5, the variational distribution over $z_0$ was computed using a feedforward network with a single hidden layer of 250 $\tanh$ units. Samples of $z_0$ were converted into initial hidden/visible states for the primary and guide LSTMs using a feedforward network with a single hidden layer of 250 $\tanh$ units. The latent variable $z_0$ was in $\mathbb{R}^{20}$ and the latent variables $z_t$ for $t > 0$ were in $\mathbb{R}^{100}$.

We trained the models using minibatches of size 250. For each example in the minibatch we sampled a single trajectory from the guide policy. The necessary KL divergences were computed via partial

Rao-Blackwellisation, i.e. at each step we computed a 1-step KL analytically, and the sum of these provided an estimator whose mean was the full-trajectory KL.

In the generative tests, we trained the "raw" model for 200k updates. The variational posterior fine-tuning stage lasted 50k updates. We used the ADAM algorithm for optimization [6], which includes both first-order momentum-like smoothing and second-order Adagrad-like rescaling. We used a learning rate 0.0002 for all models in all tests.

The imputation tests added a "reader" LSTM to the generative model (i.e. the primary policy). This had precisely the same structure as the guide LSTM. However, rather than inputting $[c_t; \hat{c}_t]$ at each step (which includes information about the target values in $x_*$), we simply input $[c_t; c_t]$. This was the first thing we tried, and it worked alright, but could probably be improved.

We used the rather new Blocks framework for managing all of our LSTM-based models, though we only really used the framework for managing the THEANO computation graph [20, 1]. All training and data management were done manually in our test scripts. In addition to the LSTM-based models, we also implemented the GPSI models and baselines using THEANO.

We trained our GPSI models using the same basic setup as for the LSTM models. For MNIST tests, the three networks underlying the model were built using two hidden layers of 1000 ReLU units. For the TFD and SVHN tests the layers were increased to 1500 units. We used latent variables $z_t \in \mathbb{R}^{100}$ for MNIST and $z_t \in \mathbb{R}^{200}$ for TFD/SVHN. Batch sizes and optimization method were the same as for the LSTMs. Code is available on Github. Due to computation/time constraints we performed little/no hyperparameter search. The GPSI results should improve somewhat with better architecture choices. Adding the localized read/write mechanisms from [4] may help too.

## 9.2 Problems with VAE Imputation

Variational autoencoder imputation proceeds by running multiple steps of iterative sampling from the approximate posterior $q(z|x)$ and then from the reconstruction distribution $p(x|z)$, with the known values replaced by their true values at each step. I.e. the missing values are repeatedly guessed based on the previous guessed values, combined with the true known values.

Consider an extreme case in which the mutual information between $z$ and $x$ in the joint distribution $p(x, z) = p(x|z)p(z)$, arising from combining $p(x|z)$ with the latent prior $p(z)$, is 0. In this case, even if the marginal over $x$, i.e. $p(x) = \sum_z p(x|z)p(z)$, is equal to the target distribution $\mathcal{D}_\mathcal{X}$, each sample of new guesses for the missing values will be sampled independently from the marginal over those values in $\mathcal{D}_\mathcal{X}$. Thus, the new guesses will be informed by neither the previous guesses nor the known part of the observation for which imputation is being performed.

In addition to this fundamental defect, the VAE approach to imputation also suffers due to the posterior inference model $q(z|x)$ lacking any prior experience with heavily perturbed observations. I.e., if all training is performed on unperturbed observations, then the response of $q(z|x)$ can not be guaranteed to remain useful when presented with observations from a different, perturbed distribution.

While one could train a basic VAE for imputation by sampling random "VAE imputation" trajectories and then backpropagating the imputation log-likelihood through those trajectories, we empirically found that this was largely ineffective. In a strong sense, the problem with this approach is analogous to that solved (in certain situations) by guided policy search. I.e., the primary policy is initially so poor that an, e.g., policy gradient approach to training it will be uninformative and ineffective. By incorporating privileged information in the guide policy, one can slowly shepherd the initially poor primary policy towards gradually improving behavior.

## 9.3 Additional Qualitative Results for GPSI Models

(a)

(b)

(c)

Figure 8: This figure illustrates roll-outs of (a) additive (b) jump, and (c) variational auto-encoder policies trained on MNIST as described in the main text. The ways in which the additive and jump policies proceed towards their final imputations are visually distinct. We ran two independent roll-outs of each policy type for each initial state, to exhibit the ability of our models to produce multimodal imputation densities. All initial states were generated by randomly occluding a 16x16 block of pixels in images taken from the validation set. I.e. these initial conditions were never experienced during training. Zoom in for best viewing.

(a)

(b)

(c)

Figure 9: This figure illustrates roll-outs of (a) additive (b) jump, and (c) variational auto-encoder policies trained on (grayscale) SVHN as described in the main text. The ways in which the additive and jump policies proceed towards their final imputations are visually distinct. We ran two independent roll-outs of each policy type for each initial state, to exhibit the ability of our models to produce multimodal imputation densities. All initial states were generated by randomly occluding an 17x17 block of pixels in images taken from the validation set. I.e. these initial conditions were never experienced during training. Zoom in for best viewing.

(a)

(b)

(c)

Figure 10: This figure illustrates roll-outs of (a) additive (b) jump, and (c) variational auto-encoder policies trained on TFD as described in the main text. The ways in which the additive and jump policies proceed towards their final imputations are visually distinct. In particular, the "strategy" pursued by the jump policy is not intuitively clear. We ran two independent roll-outs of each policy type for each initial state, to exhibit the ability of our models to produce multimodal imputation densities. All initial states were generated by randomly occluding a 25x25 block of pixels in images taken from the validation set. I.e. these initial conditions were never experienced during training. Zoom in for best viewing.