[Reviews · NeurIPS 2015]

Submitted by Assigned_Reviewer_1

The most impressive experimental result in the paper is the simple and natural extension of the DRAW model proposed in Section 2.5 which does seem to perform better than the original model. The imputation results look reasonable but it is hard to say how good they are as the baselines are rather weak.

It seems unfair to compare a variational autoencoder trained on images without any missing pixels to models trained explicitly to impute. An order-agnostic deep NADE model (A Deep and Tractable Density Estimator, Uria et al.) would be a much more reasonable baseline.

======================= In light of the authors's clarifications I increased my score to 7.
Summary: The authors reinterpret ancestral sampling in directed generative models as sequential decision making and variational training of such models as guided policy search (GPS). While this is certainly technically correct, it is not clear to me from the paper that this is a fruitful thing to do. The models developed in the paper are extensions of the DRAW model, which was formulated in the stochastic gradient variational Bayes (SGVB) framework, and are at least as natural under it as under the GPS framework. The paper would really benefit from some examples where GPS is the more natural choice.

Submitted by Assigned_Reviewer_2

The paper re-interprets

a family of generative statistical models under the framework of sequential decision making used in reinforcement learning.

It develops connections between training algorithms from the two fields. The paper first illustrates the ideas using 3 generative models and then develops a more general framework for a task called data imputation which encompasses density estimation/classification/regression. Models following this framework are tested on image datasets and compared to a baseline.

The paper proposes a new and

interesting view of generative models under the light of sequential decision making. This work clearly opens new perspectives and should allow the formulation of new ways to address learning and inference problems in a broad class of statistical models. Globally, the idea is stimulating. My main complaint concerns the form of the paper. First, the writing is extremely dense with many shortcuts, which makes it hard to follow. Second, the authors assume the reader to be totally familiar with developments concerning guided policy search introduced in a series of papers very recently (2013-2015).

Without this it is impossible to follow the technical details. The authors provide a nice intuition about the main ideas of the paper by developing examples in section 2.

It is more difficult to understand how the model introduced in section 3 really works, not talking about reproducing the implementations. The latter seem relatively complex, making use of different estimation methods briefly mentioned in the text. It is suggested to introduce some pedagogical material in the supplementary document.

The same remark holds for the description of the experiments. There is no description of the datasets, no precise description of the task; different choices are made with no justification/ explanation (the different d values, the number of imputation steps, etc). The comparison with baselines itself is weak. The authors mention that there are only few general imputation models. This might be true, but for the specific tasks handled here, there exist several other models which could be used as baselines. It is not clear for me if and how guided policies may be defined for general "imputation" problems and where the additional information used by this policy comes from.

Additional examples might help. The discussion is focused on guided policy search. Is the connection between generative modeling and reinforcement learning more general or is it restricted to this specific family of methods?
Summary: The paper introduces a new connection between generative statistical models and sequential decision making which opens new perspectives for the development and interpretation of statistical models.

The idea is very interesting. The paper is however extremely dense (models and technical implementation descriptions), and cannot be followed without a very good knowledge of recent developments by the "presumably" same authors. The experimental validation is weak.

Submitted by Assigned_Reviewer_3

The paper proposes two main contributions : the first one consists in revisiting existing generative models (i.e autoregressive networks, DRAW, and Deep unsupervised learning using nonequilibrium thermodynamics) as sequential decision processes that can be solved using Guided policy search approaches. The second contribution consists in, by considering this generic framework, extending existing generative approaches for the generic problem of data imputation by proposing two different models - with two variants for each model depending on how the generated output is built. Experiments are made on three images datasets and show the effectiveness of the proposed approaches.

Fitting different models into the same guided policy search formalism is a very interesting approach which provides a general point of view of the existing works. It particularly gives a nice way to compare the assumptions of the past and future generative models. My only concern is that Section 2 is difficult to read since it needs the reader to have a good knowledge of these different models, and also of the Guided policy search approach. But this section clearly brings an original idea which will be of the interest of many researchers. The most important contribution of Section 3 is to clearly define the data imputation problem as a sequential process that can be solved using guided policy search. Particularly, the fact that data imputation covers the problem of finding missing values, but also classification/regression problem is interesting, but not well developed in the paper - a small discussion on this point would strengthen the paper. The two models (Section 3.1 and Section 3.2) are new generative models - very close to the DRAW approach - that demonstrate the fact that the generic point of view provided in this article allow us to ''easily'' imagine new approaches. A discussion on the complexity of the models, and their ability to deal with large datasets and large number of features could be interesting. Concerning the experiments, my only concern is that they have been made only on image datasets, which are particular cases of data imputation where features share some strong relations. Using other datasets (for example collaborative filtering datasets which correspond to typical data imputation problems) would reinforce the paper.

The paper is very dense, and the definitions of the models are difficult to follow - adding some figures allowing to understand the architecture of the different methods would help the reader, but space is limited. For example, in my opinion, Figure 3 does not bring so much information and could be replaced by more informative pictures focused on the core of the paper.

Pro:

- The paper proposes a generic framework for building/understanding deep generative models - Two new models are proposed for the generic sequential data imputation problem - The article is very dense

Cons: - The article is very dense, and needs a very good knowledge of previous works - Generative models are only tested on image datasets

Summary: The contributions of the paper are very strong: a unified vision of existing models + new models. The paper is hard to follow since it is very dense.

Submitted by Assigned_Reviewer_4

This paper connects methods for policy search in reinforcement learning with inference techniques for general generative models. Several inference methods are reviewed and extended. The second part of the paper applies these to imputation, which amounts to reconstructing damaged images from e.g. the MNIST dataset.

This paper may seem very solid from a technical point of view, but at the same time it is not easy to read "story-wise". I can follow the technical steps in the paper (although I am not an expert in some aspects of the specific models), but for me many of these are described without enough "story". The introduction (and abstract) on the first page are very promising, but then the next five pages are very technical. This can be good, only if provided with enough intuition, such as is done in lines 207etc and 299-301, but there are too few of such fragments. Some of the insights (such as lines 125-126) are the key components of this paper, but they are motivated typically by looking at the formulas describing them. Problem statements such as equation 2 can be the right level of looking at similarities between problems (and solutions) but much of the remaining parts of (e.g.) sections in 2 are too detailed, IMHO. Then again some parts are very specific about upgrading specific techniques (such as 224-231) and performance comparisons, or about representations of policies (as two-layer networks with specific properties), although the paper (as described in the introduction) would be about sequential decision making for generative models (e.g. sampling).

What I basically miss is a roadmap starting from the conceptual start of identifying where policy search and general inference meet, all the way through the various models and extensions. Note that I do appreciate the paper being very technical, and very specific/precise, but for me it does not seem always at the right level. The core aspect of the paper (data generation as sequential decision making) is not very novel, since this topic has been approached before (e.g. grammars and reinforcement learning based training of them, to name one example) but the solid way this paper approaches it (in the current state-of-the-art inference techniques) seems a good way to go. The second part of the paper (section 3) does a better job by first describing an intuitive model of imputation (the problem) and then providing technical details of the solution. (I would like to see some more intuition and description of the reward component though, since that may not be obvious to all readers, including me). I might even have the feeling that imputation would be a good problem to start the paper with, as an illustration of the proposed framework. Sections 3.1. and 3.2. are then somewhat unbalanced again in which details they describe and which should be "known" to the reader from existing literature. I'd rather read more intuition again (such as 299-302), and slightly less details. The details concerning the read and write operations in section 3.2. are not well enough described to understand/appreciate what happens here.

Overall, the methods and the imputation setting are relatively clear when arriving in section 4. However, the introduction mentions that there are no alternatives to compare to, such that the authors compare internally, and against a baseline. I do not fully understand why one cannot find alternative techniques in the area of computer vision (for example Francis Bach's work, but there are others) when it comes to image reconstruction (in probabilistic/generative settings). The results seem "ok", but there's not much help to evaluate the findings. The authors state that "appear quite promising", and visually inspecting the results may verify that, but again this evaluation is not totally conclusive. At the end of section 4 it seems that performance on this task is important, whereas before it seemed that comparing the extension (of the LSTM model) against the original was important, whereas even earlier it seemed that connecting generative models and sequential decision making was important. Maybe these different goals, with different levels of description, are the weaknesses of the paper.

Overall, a solid technical approach with a description that is maybe not at the right level (of detail). The writing is clear, polished, and very precise. In addition, the experimental results are nice, and they show the viability of the approach, but they may not necessarily demonstrate how to pick which method here, and they do not necessarily exemplify the main thesis of the paper (see the title). I guess that it would be much better to transform parts of this paper to focus more on ideas and to provide a heavy technical report with all the details concerning specific Bernoulli distributions and all that. Nevertheless, even in this form it is an interesting paper.
Summary: Solid paper, but slightly too technical, which makes the paper weaker in some aspects. Experimental results can be improved (especially concerning a comparison).

Author Feedback
Author rebuttal: Thank you for the constructive and thoughtful suggestions. We respond to some general points that recurred across reviews.

The reviewers found the paper interesting but overly dense and, in some parts, over-precise. While the content requires a fairly dense presentation due to space constraints, we agree that the writing can be substantially improved. We've cleaned up the notation and presentation of Sec. 2 and 3, which improves readability and removes some redundancies. This reduces the burden on the reader and frees space for visualizations of our models. We will also include a brief overview of guided policy search and pseudo-code for our models' iterative generation and training procedures in the supplementary material. The code for our models and experiments will be available on GitHub. We will move model implementation details, like the size and type of hidden layers or choice of output density model, to the supplementary material. This will free up further space for intuitive exposition and conceptual explanation of the models, without removing any core content. We will expand and clarify the experiment descriptions.

We agree that the baselines we chose are not the strongest. We will specifically mention how, e.g., NADE, Multi-prediction Deep Boltzmann Machines, and density networks have an "imputational" form. That said, we could significantly improve the quantitative and qualitative performance of our models just by switching to task-specific architectures based on, e.g., convnets. The structure of our models makes this sort of change easy. In contrast, how to incorporate task-specific architectures in NADE is not so clear. While NADE-style models can be efficient for density evaluation, they are costly when generating samples, as they require non-trivial updates for each sampled value. In contrast, our models use a constant number of resampling steps for any number of missing values. In short, approaches based on multiple stages of sampling and/or iterative refinement offer many advantages over NADE-style models that don't use latent variables.

As for providing more informative empirical comparisons, we now have tests comparing the performance of our models when trained with different numbers of refinement steps. This allows us to treat our 1-step models as a sort of baseline. Multiple refinement steps definitely improve over a single step, though overfitting can become a problem with more steps and larger models. This could be improved in the future with better regularizers for generative models. We chose layer sizes to be large but computationally manageable. Our new tests, covering a range of step counts, should make step count selection a non-issue. For MAR, we chose the occlusion size to be large enough to make many reconstructions multi-modal.

We will add the DRAW+attention result for binary MNIST to the paper. We note that it uses a task-specific architecture.

While our new methods could also be understood in the Stochastic Gradient Variational Bayes framework, we believe that the connection to RL will prove both interesting and useful in the long run. For example, consider recent results suggesting that bad local minima are not a significant problem for large, deep networks. These results may be only weakly informative for models involving significant "stochastic decision depth". Current SGVB-style algorithms for training deep, directed generative models are all equivalent to some form of policy search. But, policy search is known to be susceptible to local minima and high variance in the required gradient estimates. This suggests that it might be worth developing alternative algorithms based on, e.g., Q-Learning or Actor-Critic architectures. It may also be worth focusing more on the role of exploration when training deep, directed generative models. These ideas are less obvious without the RL connection. While the methods present in the deep learning literature are most similar to (guided) policy search, the connections we propose are not restricted to a particular RL algorithm.

The general approach we introduce for constructing conditional distributions by iteratively refining a "working hypothesis" based on feedback from previous actions allows a straightforward way of modelling complex dependencies in the output. And, it does so in a well-grounded, fully probabilistic setting. This could benefit many structured prediction problems. E.g., one could perform per-pixel image segmentation by iteratively "rendering" the current segmentation and then refining it after comparing it to the underlying image, or one could perform object localization by iteratively "rendering" the current bounding box estimate and then refining it after comparing it to the underlying image. As suggested by one reviewer, non-image-based applications are possible as well, e.g. speech reconstruction fits our framework very well. However, experiments in other tasks go beyond the scope of the paper.